# Intra-Host Citrus Tristeza Virus Populations during Prolonged Infection Initiated by a Well-Defined Sequence Variant in *Nicotiana benthamiana*

**DOI:** 10.3390/v16091385

**Published:** 2024-08-30

**Authors:** Tathiana Ferreira Sa Antunes, José C. Huguet-Tapia, Santiago F. Elena, Svetlana Y. Folimonova

**Affiliations:** 1Department of Plant Pathology, University of Florida, Gainesville, FL 32611, USA; tathianaferreira@ufl.edu (T.F.S.A.); jhuguet@ufl.edu (J.C.H.-T.); 2Instituto de Biología Integrativa de Sistemas (I2SysBio), CSIC-Universitat de València, 46980 Valencia, Spain; santiago.elena@csic.es; 3Santa Fe Institute, Santa Fe, NM 87501, USA

**Keywords:** intra-host population diversity, defective viral genomes, citrus tristeza virus

## Abstract

Due to the error-prone nature of viral RNA-dependent RNA polymerases, the replication of RNA viruses results in a diversity of viral genomes harboring point mutations, deletions, insertions, and genome rearrangements. Citrus tristeza virus (CTV), a causal agent of diseases of economically important citrus species, shows intrinsic genetic stability. While the virus appears to have some mechanism that limits the accumulation of single-nucleotide variants, the production of defective viral genomes (DVGs) during virus infection has been reported for certain variants of CTV. The intra-host diversity generated during plant infection with variant T36 (CTV-T36) remains unclear. To address this, we analyzed the RNA species accumulated in the initially infected and systemic leaves of *Nicotiana benthamiana* plants inoculated with an infectious cDNA clone of CTV-T36, which warranted that infection was initiated by a known, well-defined sequence variant of the virus. CTV-T36 limited the accumulation of single-nucleotide mutants during infection. With that, four types of DVGs—deletions, insertions, and copy- and snap-backs—were found in all the samples, with deletions and insertions being the most common types. Hot-spots across the genome for DVG recombination and short direct sequence repeats suggest that sequence complementarity could mediate DVG formation. In conclusion, our study illustrates the formation of diverse DVGs during CTV-T36 infection. To the best of our knowledge, this is the first study that has analyzed the genetic variability and recombination of a well-defined sequence variant of CTV in an herbaceous host.

## 1. Introduction

RNA viruses are characterized by high error-prone replication, a short generation time, and large population sizes, which results in a broad cloud of closely related but genetically distinct variants coexisting within an infected host that is commonly referred to as viral quasispecies [1,2,3,4]. This standing genetic variability provides RNA viruses with a significant advantage in adapting to new environments, overcoming host responses, and potentially accelerating the emergence of new viral strains [5,6].

Typically, RNA viruses produce multiple species of RNA molecules during infection, which include full-length progeny genomes and their complementary copies plus several species of less-than-the-complete-genome size, such as subgenomic RNAs (sgRNAs) that serve as mRNAs for the translation of certain viral genes and also defective (DVGs) or non-standard viral genomes [7]. DVGs are rearranged viral genomes that are generated by many viruses upon replication. Some of these DVGs are unable to replicate independently without the help of a co-infecting full-length virus. Furthermore, in some cases, DVGs become encapsidated in a virus particle and become part of the viral population [8]. DVGs with the ability to interfere with the amplification of their parental virus and influence the viral population dynamics or modulate the host immune responses are also called defective interfering genomes [9].

DVGs and the respective parental viral genomes share at least the essential characteristics needed for replication, such as RNA structural elements present at the 5′ and 3′ termini and required for RNA-dependent RNA polymerase (RdRp) recognition and the initiation of replication, and some can harbor the sequences needed for packaging into a virion. Next-generation sequencing (NGS) has revealed a large variety of DVG species produced upon viral infections, and four main classes of DVGs have been defined: deletions, insertions, and snap-back and copy-back genomes [8,10]. The latter two types of DVGs, copy- and snap-backs, refer to the DVGs that are generated when the RdRp detaches from the template and reattaches to the nascent strand, while copying back a strand that is complementary to its own 5′ end [11]. The existence of DVGs has already been reported for more than ten plant viruses, and their generation has been shown to arise with the progression of infection and is enhanced during serial passages of the virus in the plant hosts under laboratory conditions [12].

Citrus tristeza virus (CTV) (species *Closterovirus tristezae*; genus *Closterovirus*; family *Closteroviridae*) is responsible for two of the most devastating viral diseases of citrus, quick decline and stem pitting, which have affected citrus production worldwide for more than a century [13,14,15,16]. CTV possesses the largest non-segmented RNA genome among plant viruses that is encapsidated by the major and minor coat proteins (CP and CPm, respectively), forming a long flexuous virion. The 19.3 kb, single-stranded, positive-sense RNA genome of CTV contains 12 open reading frames (ORFs) (Figure 1) [16,17]. ORFs 1a and 1b are translated directly from the genomic RNA and encode proteins that mediate virus replication [17,18]. The remaining ten ORFs are expressed from the 3′-coterminal sgRNAs. The CP, CPm, p65 or HSP70h (a homolog of the cellular proteins in the HSP70 family), and p61 are required for virion assembly and, along with a small hydrophobic protein, p6, are involved in virus movement [19]. The CP, p20, and p23 function as viral suppressors of RNA silencing (VSRs) [20]. CTV also produces three proteins, p13, p18, and p33, that are needed for the virus’s ability to infect an extended host range [21,22]. Additionally, the p33 protein plays a crucial role in mediating CTV superinfection exclusion (SIE), a phenomenon that renders a host plant infected by a CTV variant protected against a secondary infection with a variant of the same virus strain [23].

At present, at least eight strains of CTV have been recognized. The classification is based on the full-genome sequences, where variants with less than a 7.5% difference at the nucleotide level are grouped into the same strain [13,24,25]. Within the same strain, variants of CTV show a high degree of similarity. For instance, for the VT strain, the average nucleotide identity between individual variants is 96%, and this strain is considered the most diverse of all the recognized strains. The T36 strain is notably more uniform, with an average nucleotide identity of 99%. Remarkably, variants within the same CTV strain can induce a range of phenotypes, from a symptomless infection to a severe disease. Besides the genetic background of a virus variant, the symptoms induced depend on the type of host, such as the scion and rootstock of a citrus tree, and the interactions between CTV variants simultaneously infecting the same host [13].

The genome of CTV is known to have marked genetic stability [26,27,28]. However, considering the low copying fidelity of viral RNA polymerases, it is not well understood how the stability of the large genome of CTV is maintained. One possible mechanism has been described for coronaviruses, which encode a proof-reading 3′-to-5′ exoribonuclease [29]. In contrast to what has been found with coronaviruses, RNA polymerases of closteroviruses lack proof-reading exonuclease activity. It has been proposed that the phenomenon of SIE could function to maintain the genomic stability of CTV by blocking the co-replication of the progeny genomes resulting from the error-prone replication process [27]. Interestingly, the role of viral SIE in maintaining the optimal error frequency during the synthesis of viral RNA genomes was also suggested in a study with turnip crinkle virus (TCV) that demonstrated that TCV SIE occurs at the level of the incoming virus replication and is mediated by the viral replicase produced by the primary infecting virus [30]. While CTV appears to have a mechanism to limit the accumulation of single-nucleotide variants, a diversity of variants is generated through recombination [31]. The production of one or more defective RNA molecules of various sizes has been reported for several CTV isolates [32,33,34]. Most of these DVGs harbor large deletions affecting several essential genes, while retaining the 5′ and 3′ termini [32,33,35,36]. Interestingly, previous data suggest that DVGs appear to be less common for the T36 strain [34].

The experimental evolution of plant viruses is typically assessed by comparing the genetic sequences of the initial virus population used for the inoculation of the first plant with those of the virus population after it has been propagated in the host for a period of time. The addition of serial passages, consisting of the horizontal plant-to-plant transmission of replicating viral populations, could promote the evolution of the virus population and, thus, the adaptation to the host. In most such experiments, the first infection is achieved using a known single variant of the virus, such as a cDNA clone or its RNA transcript. Sequential inoculations are made via mechanical inoculation using an extract obtained by homogenizing virus-infected tissue from a plant infected earlier than would be expected to harbor a heterogeneous viral quasispecies generated during infection in the plant [37,38]. This approach is quite challenging in the case of CTV for several reasons. First, CTV is a phloem-limited virus. Until now, all attempts to inoculate any of the CTV hosts following the procedures routinely used with most plant RNA viruses, such as rub inoculating an RNA transcript corresponding to the viral genomic RNA or a virus suspension onto the leaves, have been unsuccessful [16]. A standard procedure of inoculating citrus plants with CTV involves the application of a virus suspension obtained from an infected plant (a citrus or *Nicotiana benthamiana* plant) or grafting a piece of bark derived from an infected citrus plant. With this, infection of CTV in citrus plants cannot be initiated with a single-sequence variant and rather starts with a sample of the viral quasispecies accumulated in the source plant. Furthermore, infection in citrus takes months, so plant-to-plant passages would take years.

In this study, we took advantage of an herbaceous CTV host, *N. benthamiana*, which could be inoculated using the *Agrobacterium*-mediated infiltration of a plasmid harboring a viral cDNA clone. This ensures that infection is initiated by a well-defined sequence variant. To assess the genetic diversity of the intra-host populations formed upon systemic virus infection, we allowed the virus to propagate in the infected plants over a period of two months, which was followed by the deep sequencing of viral RNA species. In agreement with previous studies, the analysis of the RNA species accumulated in *N. benthamiana* plants inoculated with a cDNA clone of the T36 variant of CTV (CTV-T36) demonstrated that CTV-T36 limits the accumulation of single-nucleotide variants. CTV-T36 also produced DVGs that contained insertions and internal deletions and underwent recombination at distinct sites across the genome, which were present as minor components of the viral populations. Evaluation of next-generation RNA sequencing data from the initially inoculated leaves and systemic leaves revealed many hot-spots for recombination in specific viral genome segments. Further, short direct sequence repeats were identified at some junction sites, indicating that DVG formation could be directed by sequence complementarity. Additional investigation is needed to experimentally validate the existence of these DVGs. To the best of our knowledge, this is the first study that has analyzed the genetic variability and recombination of CTV upon the infection of an herbaceous host initiated with a well-defined sequence variant of the virus.

## 2. Materials and Methods

### 2.1. Agroinfiltration of a Virus Construct into N. benthamiana Leaves, Examination of the Results of Virus Inoculation, and Sample Collection

A recombinant plasmid vector, pCAMBIA-1380, carrying a full-length cDNA clone of the CTV-T36 variant tagged with a green fluorescent protein (GFP) gene and placed under the cauliflower mosaic virus 35S promoter (pCTV-GFP) was described earlier [39]. Agroinfiltration was conducted as previously described [39,40]. Briefly, pCTV-GFP plasmid was introduced by heat shock into *Agrobacterium tumefaciens* EHA105, and the resulting transformant was selected on the Luria–Bertani plate, containing 50 mg/mL rifampicin and 25 mg/mL kanamycin. One single colony was grown overnight at 28 °C, and the cells were gently resuspended in a buffer containing 10 mM 2-(N-morpholino)ethanesulfonic acid (MES, pH 5.85), 10 mM MgCl_2_, and 150 mM acetosyringone at O.D._600nm_ = 0.1. After two hours of incubation at room temperature and without shaking, the suspension was infiltrated into five biological replicates of six-week-old *N. benthamiana* plants using a needleless syringe. The inoculated plants were kept in a growth room under a 14 h light regime and a temperature of 22 °C for two months, during which the plants were monitored for the development of virus infection by observation of the GFP fluorescence. The GFP fluorescence in the inoculated leaves and systemic leaves of *N. benthamiana* plants was observed using a hand-held ultraviolet (UV) lamp (365 nm, UVP, Upland, CA, USA) set in a dark room. The infiltrated leaves were sampled at two weeks post infiltration (wpi) (three plants in total). At eight wpi, the same plants were used to collect leaf tissue from the upper leaves, which became infected and showed comparable GFP accumulation.

### 2.2. RNA Extraction, Library Preparation, and High-Throughput Sequencing

Total RNA was extracted from 150 mg of tissue harvested from three inoculated leaves and three upper leaves that became infected as a result of the systemic spread of the virus (the latter leaves are referred to hereafter as “systemically infected leaves”) using the Direct-zol RNA MiniPrep Plus kit (Zymo Research Corporation, Irvine, CA, USA), following the manufacturer’s instructions, and eluted in 50 μL of nuclease-free water. The RNA profile and quality were analyzed using Agilent Technologies Tapestation. The RNA-seq library was constructed using the TruSeq Stranded Total RNA with Ribo-Zero Plant (Illumina, Inc., San Diego, CA, USA). The size of the PCR-enriched fragments was confirmed by checking the template size distribution obtained on an Agilent TapeStation 4200 (Agilent, Santa Clara, CA, USA). The library was sequenced on the NovaSeq6000 S4 flowcell with 150 bp paired-end reads, generating per sample a total output of more than 20 million reads (Table 1; IL, inoculated leaves; SL, systemically infected leaves).

### 2.3. Genome Assembly and Single-Nucleotide Polymorphism (SNP) Calling

Total RNA-seq reads were analyzed using FastQC [41]. The raw reads were filtered and trimmed using Trim Galore [42], which allowed us to remove adapters and trim low-quality bases (Phred score > 30). For viral genome SNP calling, trimmed reads were mapped to the reference sequence of the infectious clone CTV-T36 tagged with the GFP using Bowtie2 [43]. Resulting SAM files were BAM-converted, sorted, indexed, and analyzed with SAMtools [44]. To calculate the average coverage for the CTV genome, the total number of reads at 20 positions between the positions 1000 and 20,000, in intervals of 1000 nucleotides, were summed and then divided by 20. SNP calling was performed using LoFreq [45]. We considered as true variants all those with a frequency > 1%, with a minimal coverage of 3000.

### 2.4. Defective Viral Genome (DVG) Identification 

The detection of DVG reads was performed using the metasearch tool DVGfinder [46]. DVGfinder integrates two of the most used DVG search algorithms, ViReMa-a [47] and DI-tector [48], into a single workflow. The trimmed reads were first mapped to the genome of *N. benthamiana* (Nbv0.5.genome.fa) using Bowtie2 [49] to remove the host reads. The remaining unmapped reads were interleaved in a unique fastq file, and this file was used to identify recombination events. The identified DVGs were characterized by three parameters: breaking point (BP), a virus genome position at which the RdRp is released from the template; rejoining point (RP), a genome position at which the RdRp reattaches to the template and continues polymerizing; and the sense of the fragments pre- and post-BP/RP junction.

## 3. Results and Discussion

### 3.1. Intra-Host Evolution of CTV-T36 and Comparison with Other RNA Viruses

Six samples were successfully sequenced: three samples from the inoculated leaves collected at two wpi and three samples from the systemically infected leaves collected at eight wpi. The average depth of the sequencing coverage of the CTV genome was approximately 7000-fold (ranging between 4453- and 16,324-fold) for the inoculated leaves and 13,000-fold (ranging between 7555- and 45,205-fold) for the systemically infected leaves, with reduced coverage at the extreme 5′ and 3′ termini. In total, we observed six SNPs: four were synonymous and two were nonsynonymous (small nonpolar by small polar). All mutations were found in minor variants and occurred in less than 5% of the reads, and all were found in a single sample only (Table 2). No SNPs were reported for two samples from the inoculated leaves and for one sample from the systemically infected leaves.

It is relevant to compare the low SNP variability found here with the CTV-infected samples with what has been shown in other studies on viral intra-host diversity. For example, Dunham et al. (2014) described the variability of zucchini yellow mosaic virus within a single *Cucurbita pepo* plant, conducting a sequence analysis of 23 infected leaves growing sequentially along the vine. Out of 112 virus variants identified across the dataset, 22 were found in multiple leaves, and 3 of the 13 mutations occurring in the virus variants in the inoculated leaves were found in the subsequent leaves in the vine. Thus, despite population bottlenecks within the host during systemic movement, multiple variants were detected and translocated between leaves [50].

The molecular evolution of tobacco etch virus (TEV) was explored in its natural host, *Nicotiana tabacum*, and in an alternative host, *Capsicum annuum* [51]. In the study, considering all the passages and leaves sequenced, 101 SNPs in *C. annuum* and 229 SNPs in *N. tabacum* were detected. Out of these, 36 SNPs were unique in *C. annuum* and 101 in *N. tabacum*. This result indicates that the dynamics of molecular evolution are different between hosts. Furthermore, the accumulation of neutral mutations was higher in the natural host, while mutations under positive selection were pervasive in the novel host [51]. To compare these results with those obtained in our study, we looked at the reported TEV diversity in the leaf samples collected after the first passage. For the inoculated leaves in *C. annuum*, seven and two SNPs were detected in each replicate, while in the systemic leaves, nine and five SNPs were found. This result indicates that the CTV intra-host diversity in *N. benthamiana*, a non-natural host, seems lower, compared to that observed for TEV.

Earlier, the intra-host evolution of CTV-T36 was assessed through the sequencing of a population of small RNAs extracted from a *Citrus macrophylla* plant inoculated with a virion preparation obtained after serial protoplast passages of the virus, with the initial protoplast batch inoculated with an RNA transcript of the CTV cDNA clone and cultivated in a greenhouse for seven years. The analysis demonstrated that, during this time, only nine nucleotide changes across the full virus genome were fixed in the population, and the mutation rate was calculated as 6.67 × 10^−5^ mutations per site and per year [24]. Due to such a low evolutionary rate, it would be unlikely to observe a nucleotide substitution over a two-month period, the time frame used in our study. This agrees with the results shown here, as we could not identify any fixed substitution during the experiment. Thus, regardless of the host, CTV exhibits a low mutation rate.

The results obtained in our work with an herbaceous host, *N. benthamiana*, are in agreement with the outcomes of the study that analyzed viral populations in the citrus host and demonstrated that CTV has a lower mutation rate during systemic infection, compared to other RNA viruses. As discussed earlier, it has been proposed that CTV could use an SIE mechanism to maintain the stability of its genome [27,52]. Further research is needed to unravel how CTV limits the accumulation of single-nucleotide variants during infection in the plant hosts and the role of SIE in this process. 

### 3.2. Generation of DVGs during Experimental Evolution of CTV-T36

The same dataset used for exploring the SNPs produced during the intra-host evolution of CTV-T36 in *N. benthamiana* was used to identify recombination events. We aligned high-quality reads to the reference sequence of the CTV-T36 infectious clone using DVGfinder [46]. It is well known that most of these algorithms introduce in silico artifacts (e.g., false positives and false negatives) during their identification and assembly process. The introduction of in silico artifacts by DI-tector has been previously reported and discussed [11]. Likewise, the low sensitivity of ViReMa-a identifying copy-backs has already been mentioned [48]. DVGfinder [46] applies a gradient-boosting-classifier machine learning algorithm to reduce the number of false-positive events identified by DI-tector and ViReMa-a [46]. DVGfinder improves the sensitivity for low-coverage data and DI-tector precision for high-coverage samples [46]. Bosma et al. (2019) showed that DVGs identified by DVG-profiler at very low levels, such as four sequencing reads, were still detectable by RT-PCR using the appropriate primers [11]. Considering these results, we decided to filter the results obtained using DVGfinder and analyze only DVGs detected by both ViReMa-a [47] and DI-tector [48] or detected with at least five reads when identified by only one algorithm. The read numbers for the identified DVGs are provided in Appendix A.

In accordance with previous reports, a recombination junction is defined as a deletion or insertion greater than five base pairs [11]. CTV-T36 produced a diverse population of variants with different recombination junctions. The four types of DVGs were found in all samples (Appendix A). In the inoculated leaves, 264 DVGs were identified, of which 30 were copy- and snap-backs, 75 were deletions, and 159 were insertions. In the systemically infected leaves, 253 DVGs were identified, of which 6 were copy- and snap-backs, 68 were deletions, and 179 were insertions. Furthermore, 50 DVGs were common between samples from the inoculated leaves and systemically infected leaves.

In the inoculated leaves, 17 DVGs were common among the three samples, while 66, 65, and 32 DVGs were found exclusively in IL1, IL2, and IL3, respectively (Figure 2A). In the systemically infected leaves, 19 DVGs were common among the three samples, while 23, 50, and 89 DVGs were found exclusively in SL1, SL2, and SL3, respectively (Figure 2B). Significant differences in the distribution of DVGs were detected between the inoculated and systemic leaves (homogeneity test: *X*^2^ = 58.791, 6 d.f., *p* < 0.001).

As documented in Appendix A and summarized in Table 3, multiple DVGs with deletions and insertions were identified. In the inoculated leaves, four deletions and three insertions were the most abundant of these, while in the systemically infected leaves, three deletions and five insertions were the most frequent. The DVGs were classified according to the DVG type; BP, the genome position at which the RdRp is released from the template; and RP, the genome position at which the RdRp is reattached to the template and continues the polymerization. The DVG with BP 7020/RP 7069 was found in five out of six samples from the inoculated and systemically infected leaves. The DVGs with BP 19,211/RP 19,131 and BP 19,811/RP 19,772 were found in all samples from the inoculated and systemically infected leaves (Table 3).

To characterize the recombination sites of DVGs, we graphed the BPs and RPs of all identified DVGs from the inoculated leaves and systemically infected leaves. As shown previously, in some cases, rearrangements of the virus genome occur at favored sites, with RNA secondary structures or direct repeat sequences known as recombination hot-spots [53]. A hot-spot was defined as a clustering of recombination junction sites with high concentrations of BPs and RPs (Figure 3). Interestingly, one hot-spot-enriched region indicated as junction area A (the red box in Figure 3) was observed in all samples from the inoculated and systemically infected leaves. The respective region of the genome of the virus variant used in this study corresponds to the GFP gene inserted into the CTV-T36 sequence as a marker gene (nt positions 19,114–19,833). In another study, Kautz et al. (2020) found that the GFP gene inserted into the Venezuelan equine encephalitis virus genome was deleted slowly but persistently over sequential virus passages. This deletion was specific to the GFP gene and occurred in a progressive manner, starting with smaller segments and gradually increasing in size [54]. In our analysis, we found few DVGs with BP and RP positions inside the GFP gene (Table 3). These DVGs showed deletions of small fragments within the respective marker gene. These findings are in agreement with our earlier observations from GFP-tagged CTV variants cultivated for months in greenhouse citrus plants, some of which showed the partial or complete loss of the added foreign gene [55].

Besides the hot-spots discussed above, we found clustering of the BPs within ORF 1a and the p23 gene in nine and three of the twenty most frequent DVGs, respectively, while the majority of the RPs clustered in the 3′ region comprised the p20, p23, and GFP genes (Table 4).

To investigate whether sequence homology plays a role in DVG formation, we examined sequences upstream and downstream from the BP and RP sites of the 20 most frequent recombination junctions detected in the DVGs with deletions (Table 4). Nine recombination junctions showed sequences of at least four nucleotides that were identical in both the BP and RP sequences (Table 4). These observations suggest that RNA sequence homology could play a role in CTV recombination to generate some DVGs.

To test whether CTV-T36 DVGs exhibit sequence preferences, we identified and quantified the upstream and downstream sequences surrounding the junction BP and RP positions (Figure 4). To avoid potential biases for overrepresented DVGs, junctions were represented by one sequence and by not considering the read depth. The percentage of nucleotides was calculated and plotted for each position in a ten-nucleotide window flanking the junction site (Figure 4). Our analysis showed no strong nucleotide preference at the recombination sites in the CTV-T36 genome. Interestingly, the graph indicates a slight enrichment for both start and stop sites, with the percentage of U and A larger than 50%, compared to that of C and G.

Earlier studies have demonstrated that, upon plant infection, CTV produces numerous defective RNAs (dRNAs). Most of these dRNA molecules consist of two fragments corresponding to the 5′ and 3′ genomic regions, while having the deletions of large sequences in the internal portion of the virus genome [32,33,35,36]. CTV dRNAs have been grouped into six different classes, considering the region deleted from the CTV genome and the size of the deletions [34]. The CTV dRNAs of Class 1 are characterized by the possession of distinct 5′ and 3′ sequences of varying sizes. Some junction sites of dRNAs are flanked by direct repeats of four to five nucleotides, supporting the possibility that they are generated via a template-switching type of recombination driven by viral replicase [35]. According to our analysis, the DVGs with BP 19,70/RP 18,804 and BP 1793/RP 18,416 are similar in size to the ones identified previously for the Israeli variant of the VT strain of CTV (CTV-VT) [32,33]. It is important to mention that the infectious cDNA clone of the T36 variant (CTV-T36) used in this work has an extra GFP gene at the 3′ terminus upstream of the 3′ untranslated region, the size of which is approximately 700 nucleotides; thus, it is expected that the dRNAs found in our work are slightly bigger than those identified in previous studies. Interestingly, these DVGs showed direct repeats of the nucleotide sequences surrounding the recombination junctions, suggesting that sequence complementarity could mediate DVG formation (Appendix A).

The dRNAs of Class 2 have a common 3′ end fused to different size regions from the 5′ terminus. Previous research identified a double-stranded RNA with characteristics corresponding to the sgRNA of ORF 11 [36]. It is possible that the 5′ end of the ORF 11 sgRNA might serve as a highly specific hot-spot for RNA recombination [36,56]. In our analysis, we found DVGs with BP 232/RP 18,356, BP 394/RP 18,355, BP 490/RP 18,353, and BP 573/RP 18,355, which have the same RP position and different BP positions (Table 3). Remarkably, the sequence at the RP position is the same shown for the dRNA identified in the CTV-VT-infected bark tissue [36].

The Class 3 dRNAs have a large 5′ fragment of the genomic RNA corresponding to ORF 1a fused to different 3′ portions, some of which have a truncated ORF 10 or ORF 11. These dRNAs are encapsidated and infectious when mechanically transmitted to citrus plants or transfected into *N. benthamiana* protoplasts [57]. In our analysis, we found some DVGs with BP positions downstream of ORF 1a and RP positions inside ORF 10 and ORF 11. Although the recombination junctions of these DVGs are different from the ones identified in [57], some have repeats of four–five nucleotides near their junction sites. Thus, DVGs with BP 10,952/RP 17,835, BP 11,065/RP 19,577, BP 11,146/RP 17,950, BP 11,440/RP 19,742, BP 11,463/RP 17,924, BP 11,558/RP 17,808, and BP 11,624/RP 18,240 (Appendix A and Table 3) with similar sizes and BP and RP points could be classified as belonging to Class 3 of CTV dRNAs [57].

Class 4 of CTV dRNAs is represented by large dRNAs (LdRNAs), with different 5′ portions and all or most of the 3′ ORFs. In association with these LdRNAs, researchers have found double-recombinant dRNAs with sizes ranging from 1.7 to 5.1 kb (Class 5) that contain a non-contiguous internal sequence (generated by more than one deletion) or a non-viral sequence (generated by at least one deletion and one insertion, in addition to the two ends) [58]. Interestingly, these dRNAs identified in different CTV isolates contain an identical 948-nt 5′ region. In our study, we found three DVGs with BPs localized near position 948, including DVGs with BP 949/RP 18,355, BP 959/RP 18,350, and BP 959/RP 7325 (Appendix A). These DVGs have bigger deletions than the dRNAs identified previously; however, we cannot exclude the possibility that these DVGs were generated by two or more consecutive recombination events resulting in such deletions.

Finally, the dRNAs of Class 6 have diverse 5′ and 3′ regions and short insertions (14–17 nucleotides long) with no sequence homology with the CTV genome [32,33]. Since DVGfinder operates under the assumptions that a recombination event results in a single abnormal junction, and that once the RdRp reattaches to the template, it continues polymerizing until it reaches the end of the genome, it could not identify DVGs generated from more than one deletion or a deletion and an insertion occurring in the same genomic molecule [46].

To summarize, our results suggest that CTV possesses a mechanism to limit the accumulation of mutant variants with point mutations. Considering the large size of the CTV genome, the accumulation of mutations can be expected, and the recombination may help in preserving functional genomes, even if many non-functional recombinants are produced [59,60]. The identification of several DVGs in the leaves inoculated and systemically infected with CTV-T36 supports this possibility. Further research is needed to fully understand the mechanism of CTV recombination and how frequently DVGs are generated and how their amplification is regulated as well as their functions in the virus evolution and biology.

## Figures and Tables

**Figure 1 viruses-16-01385-f001:**
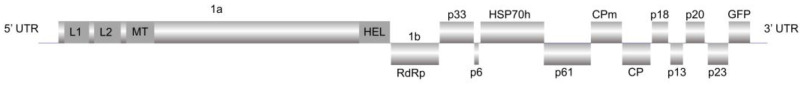
Schematic diagram of the CTV-T36 genome organization. The 5′- and 3′-untranslated regions (UTRs) along with open reading frames (ORFs) and their translation products are indicated. ORFs 1a and 1b encode several domains: papain-like protease domains—L1 (nucleotide positions (nts) 108 to 1565) and L2 (nts s1566 to 3035); MT, methyltransferase domain (nts 3222 to 4022); HEL, helicase (nts 8205 to 9398); RdRp, RNA-dependent RNA polymerase (nts 9532 to 10,851). The ten ORFs expressed from 3′-coterminal sgRNAs encode the p33 protein (nts 10,900 to 11,811); p6, a small hydrophobic protein (nts 11,885 to 12,040); p65 or HSP70h, HSP70 homolog (nts 12,046 to 13,830); the p61 protein (nts 13,754 to 15,361); CPm, minor coat protein (nts 15,336 to 16,058); CP, major coat protein (nts 16,152 to 16,823); the p18 protein (nts 16,789 to 17,292); the p13 protein (nts 17326 to 17685); the p20 protein (nts 17,761 to 18,309); the p23 protein (nts 18,391 to 19,020). The CTV-T36 infectious clone was tagged with the green fluorescent protein (GFP) ORF (nts 19,114 to 19,833).

**Figure 2 viruses-16-01385-f002:**
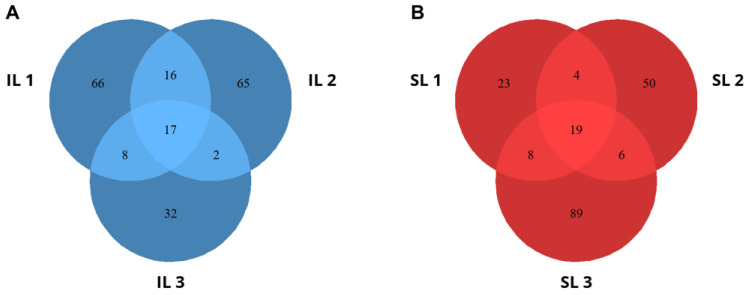
Venn diagrams presenting the numbers of DVGs detected in the samples from the inoculated and systemically infected leaves. (**A**) Venn diagram presenting the numbers of DVGs that were unique in the inoculated leaves (IL1, IL2, and IL3), with the overlapping areas showing the numbers of DVGs that were identified in all samples from the inoculated leaves. (**B**) Venn diagram presenting the numbers of DVGs that were unique in the systemically infected leaves (SL1, SL2, and SL3), with the overlapping areas showing the numbers of DVGs that were identified in all samples from the systemically infected samples.

**Figure 3 viruses-16-01385-f003:**
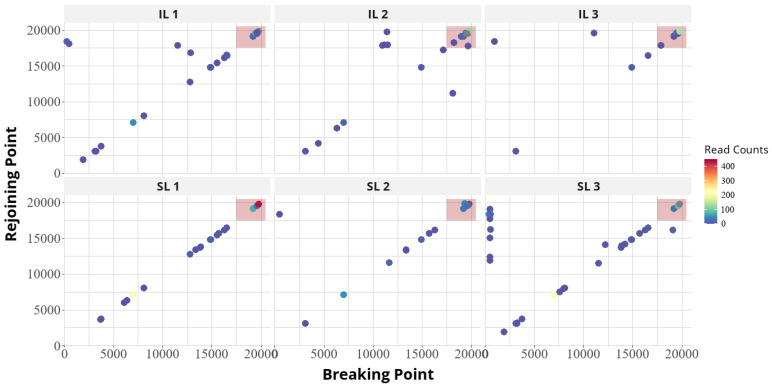
Graphs showing recombination junction site locations and read counts in sequenced CTV-T36 RNA samples. Breaking point (BP) and rejoining point (RP) distributions for DVGs detected in samples from the inoculated and systemically infected leaves. BPs and RPs are shown relative to the nucleotide positions in the virus genome. Read count is indicated by dot color, according to the legend to the right of each image. Junction maps for independently sequenced RNA preparations are shown in separate graphs labeled as IL1, IL2, and IL3 for the inoculated leaves and as S1, S2, and S3 for the systemically infected leaves.

**Figure 4 viruses-16-01385-f004:**
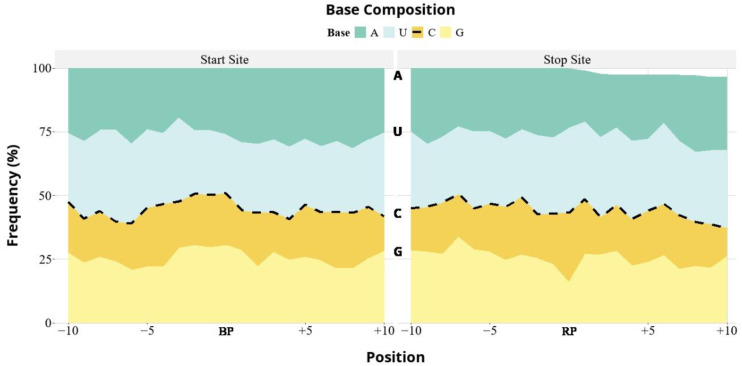
Junction site composition of the CTV-T36 DVGs. Nucleotide composition was quantified from breaking point (BP) and rejoining point (RP). Nucleotide composition was calculated as the percent of each nucleotide at each position in a 10 bp region flanking the DVG breaking and rejoining sites. The nucleotide positions upstream (−10 to −1) and downstream (+1 to +10) of the junction position are indicated.

**Table 1 viruses-16-01385-t001:** Raw data statistics for all samples used in this study.

Sample ID	Total Read Bases (bp)	Total Reads
88-1-IL1	4,167,587,618	27,599,918
88-1-IL2	4,636,013,778	30,702,078
88-1-IL3	3,592,283,960	23,789,960
88-1-SL1	4,090,091,398	27,086,698
88-1-SL2	3,609,980,254	23,907,154
88-1-SL3	3,678,504,658	24,360,958

**Table 2 viruses-16-01385-t002:** Single-nucleotide variants found across the CTV-T36-infected leaves. SNPs are shown for the samples collected at two wpi (inoculated leaves, IL) and the samples collected at eight wpi (systemically infected leaves, SL) of *N. benthamiana* plants. The reference (Ref. nt) indicates the nucleotide (nt) position in the reference CTV genome. The alternative (Alt. nt) refers to any nucleotide other than that in the reference genome that was found at the indicated position.

Sample	Genomic Position	Ref. nt	Alt. nt	Coverage	Amino Acid Change	Variant Frequency (%)
IL1	17,035	G	U	8073	Ala—Ser	1.6
SL2	4733	G	A	2724	-	1.0
SL2	15,389	G	A	3439	-	7.0
SL2	17,494	G	U	6736	Ala—Ser	1.1
SL3	3548	G	U	6520	-	1.0
SL3	15,755	G	U	8751	-	1.7

**Table 3 viruses-16-01385-t003:** DVG genomes identified in the samples from inoculated leaves and systemically infected leaves *.

	Sample	DVG_Type	BP	RP
Inoculated leaves	IL1/IL2/IL3	Deletion	19,462	19,580
	IL1/IL2/IL3	Insertion	19,211	19,131
	IL1/IL2/IL3	Insertion	19,811	19,772
	IL1/IL2	Deletion	7020	7069
	IL1/IL3	Insertion	19,204	19,124
	IL2	Deletion	18,971	19,107
	IL3	Deletion	11,065	19,577
Systemically infected leaves	SL1/SL2/SL3	Deletion	7020	7069
	SL1/SL2/SL3	Insertion	19,211	19,131
	SL1/SL2/SL3	Insertion	19,811	19,772
	SL1/SL3	Insertion	19,597	19,520
	SL1	Insertion	19,638	19,519
	SL2	Deletion	19,347	19,805
	SL2	Insertion	19,204	19,124
	SL3	Deletion	490	18,353

* Listed are the most frequently detected DVGs (based on read count) found in each group of inoculated and systemically infected leaves. The DVGs are described according the DVG_type, type of DVG; breaking point (BP), the genome position at which the RdRp is released from the template; rejoining point (RP), the genome position at which the RdRp reattaches to the template and continues the polymerization.

**Table 4 viruses-16-01385-t004:** Nucleotide sequences surrounding recombination junctions of deletion-type DVGs generated upon CTV-T36 infection *.

	BP	BP Sequence	RP	RP Sequence	DVG Length
1	232 (ORF 1a)	TCTGTTAGAATCACCAAGGTG	18,356 (btw p20/p23)	TAACTTTAATTCGAACAAATA	1983
2	394 (ORF 1a)	CCCGTTATCAACGCATCTGGC	18,355 (btw p20/p23)	CTAACTTTAATTCGAACAAAT	2146
3	490 (ORF 1a)	AGGTCCCTCCGTCAGGCAAAG	18,049 (p20)	TCGCGACAAGCTGCTCTGTAC	2548
4	490 (ORF 1a)	AGGTCCCTCCGTCAGGCAAAG	18,353 (btw p20/p23)	AACTAACTTTAATTCGAACAA	2244
5	490 (ORF 1a)	AGGTCCCTCCGTCAGGCAAAG	17,713 (btw p13/p20)	GTCTATTAGTATAACGTATTA	2884
6	490 (ORF 1a)	AGGTCCCTCCGTCAGGCAAAG	19,077 (btw p23/GFP)	AAGGGTCGTTAATTGACGACT	1520
7	518 (ORF 1a)	CTGTTTCCCTTTCTAGCCGGG	16,218 (CP)	CGATGTTGTTGCTGCCGAGTC	4407
8	573 (ORF 1a)	CACACGTTCAAGACTTCACAG	18,355 (btw p20/p23)	CTAACTTTAATTCGAACAAAT	2325
9	7020 (ORF 1a)	GAAGTTGTTACGCAACCTATT	7069 (ORF 1a)	GGGTTGTTACTTCAAACCCTT	20,058
10	11,065 (p33)	TTATTTCTCATTGTATTTCTT	19,577 (GFP)	TACATCACGGCAGACAAACAA	11,595
11	15,008 (p61)	ACGTCATAGTGAAGTGGCTTT	489 (ORF 1a)	GAGGTCCCTCCGTCAGGCAAA	5588
12	16,142 (btw CPm/CP)	ACATTTACTAGGTTTGAATTA	16,043 (CPm)	TACGCGATTTGGGTAAGTACT	20,008
13	18,530 (p23)	ATTATTATCGATGCTTTGATA	679 (ORF 1a)	TCTCACCTCCCTTACATGGGG	2256
14	18,549 (p23)	TACGGAAGAATAGTTATCAGG	18,542 (p23)	GCTTTGATACGGAAGAATAGT	20,100
15	18,971 (p23)	GAGTATCCAGTGAGTCTGAGT	19,107 (btw p23/GFP)	TTTACTAGGTTTGAATTATGG	19,971
16	19,119 (GFP)	GAATTATGGCTAGCAAAGGAG	16,156 (CP)	TGAATTATGGACGACGAAACA	17,144
17	19,347 (GFP)	ATCCGGATCATATGAAACGGC	19,805 (GFP)	GCTGGGATTACACATGGCATG	19,649
18	19,462 (GFP)	CAAGTTTGAAGGTGATACCCT	19,580 (GFP)	ATCACGGCAGACAAACAAAAG	19,989
19	19,578 (GFP)	ACATCACGGCAGACAAACAAA	19,493 (GFP)	ATCGAGTTAAAAGGTATTGAT	20,022
20	19,805 (GFP)	GCTGGGATTACACATGGCATG	19,347 (GFP)	ATCCGGATCATATGAAACGGC	19,649

* The table lists the 20 most frequently found DVGs (based on read count) with deletions, which were detected in the inoculated and systemically infected leaves. The DVGs are described according to the nucleotide positions of the breaking point (BP), the genome position at which the RdRp is released from the template (the respective ORF is indicated), and the rejoining point (RP), the genome position at which the RdRp reattaches to the template and continues the polymerization. The “btw” abbreviation indicates an intergenic region between the indicated genes. Sequences colored in red indicate regions of identical/similar sequences between the regions surrounding the BP and RP.

## Data Availability

Data are contained within the article and Appendix A.

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
