# Peer review of "Intra-Host Citrus Tristeza Virus Populations during Prolonged Infection Initiated by a Well-Defined Sequence Variant in Nicotiana benthamiana"

_viruses, 2024, doi:10.3390/v16091385_

Round 1

Reviewer 1 Report

Comments and Suggestions for Authors

Dear authors

The article is very interesting with original results in accordance with the available literature. It is very well written and experimentation conducted with appropriate tools of analysis. The formation of defective RNA during virus replication is an important aspect to be investigated in the future to understand the interaction of CTV with different host. However, it has to be considered an initial approach useful as starting point to understand the evolution of RNAs population during infection.  In fact the use of a cDNA infective clone, as well as a non host like N.bentamiana, could mimic in a perfect way the CTV infection but further confirmation are needed with a real isolate inoculated in a specific citrus host.

The article is very interesting  and original. Results are in accordance with literature described for CTV or other viruses. Experiments are well conducted and the article is very well written, clear in the language, tables and figures.

In my opinion it is ready to be published and only minor corrections are requested.

Line 126 I think you mean ”plant-to-plant”

Line 173. About the “three leaves systemically infected”..do you mean they were not directly inoculated but resulted to be infected after?

Lines 223-225. How can you explain the higher average depth of sequencing coverage in the systemically infected leaves? If possible, I  suggest to insert a comment.

Lines 347-347. About the hot-spots you have found. Have they the same sequence in all the samples? Can you explain better if they are located in a specific gene or part of the genome? And if this can have an influence in the mechanism of RNA silencing?

Line 409. I thing you missed “BP” in the “DVGs 10952/RP 17835”

Line 611. Please add the spaces between words.

Author Response

Reviewer's comments and authors' response:

Dear authors

The article is very interesting with original results in accordance with the available literature. It is very well written and experimentation conducted with appropriate tools of analysis. The formation of defective RNA during virus replication is an important aspect to be investigated in the future to understand the interaction of CTV with different host. However, it has to be considered an initial approach useful as starting point to understand the evolution of RNAs population during infection.  In fact the use of a cDNA infective clone, as well as a non host like N.bentamiana, could mimic in a perfect way the CTV infection but further confirmation are needed with a real isolate inoculated in a specific citrus host.

The article is very interesting  and original. Results are in accordance with literature described for CTV or other viruses. Experiments are well conducted and the article is very well written, clear in the language, tables and figures.

In my opinion it is ready to be published and only minor corrections are requested.

We greatly appreciate the reviewer’s comments and believe that those helped to improve the manuscript. We revised the manuscript according to the reviewer’s suggestions.

Line 126 I think you mean ”plant-to-plant”

Corrected as suggested

Line 173. About the “three leaves systemically infected”..do you mean they were not directly inoculated but resulted to be infected after?

We added some clarification regarding the systemically infected leaves in the manuscript text as follows: “…and three upper leaves that became infected as a result of the systemic spread of the virus (the latter leaves are referred here as ‘systemically infected leaves’)…” (see lines 188-189 in the revised version)

Lines 223-225. How can you explain the higher average depth of sequencing coverage in the systemically infected leaves? If possible, I  suggest to insert a comment.

We thank the reviewer for the question. Perhaps, the depth of coverage correlated with the level of virus accumulation. However, such situation (the average coverage being higher in the systemically infected leaves) is not universal, and it appears to vary between samples. For instance, in one sample from the inoculated leaf (IL1), the number of viral reads was ca. 9 million, while in sample SL2, the number of viral reads was ca. 7 million. Therefore, we think that we could not generalize this observation in a more general comment.

Lines 347-347. About the hot-spots you have found. Have they the same sequence in all the samples? Can you explain better if they are located in a specific gene or part of the genome? And if this can have an influence in the mechanism of RNA silencing?

We greatly appreciate the reviewer’s comments and believe that those helped to improve the manuscript. The BP and RP sequence for hotspots is not the same. The recombination junctions usually showed sequences of at least four nucleotides that were identical in both the BP and RP sequence. We added some clarification regarding the BP and RP positions in the CTV genome in the paragraph on lines 432-449 and 460-461.

Line 409. I thing you missed “BP” in the “DVGs 10952/RP 17835”

Corrected

Line 611. Please add the spaces between words.

Done

Reviewer 2 Report

Comments and Suggestions for Authors

The paper by Antones and colleagues is dedicated to the challenging problem of how an RNA virus genome evolves and what are the primary mechanisms that drive the genetic changes. The authors used the cDNA copy of Citrus tristeza virus to inoculate N. benthamiana host and analyzed the DVG progeny of the master sequence – an approach that seems quite logical and adequate. This allowed to obtain novel data on the copy-choice recombination events upon CTV RNA replication and the hot spots for such events. The manuscript is well written and deserves publication is Viruses.
One point which might indicate a direction for the further research. It is known – as mentioned in this paper – that CTV has marked genetic stability despite the apparent luck of the proofreading mechanism described for replicases of large nidoviruses. However, the size of the closterovirus replicase genes is twice that of those in their distant relatives belonging to the Alpha-like virus superfamily. Is it possible that the unique protein domains in the expanded replicases of closterovirids influence the recombination pattern and the observed stability of their RNAs? Hopefully, these questions will find answers in future research.

Author Response

Reviewer's comments:

The paper by Antones and colleagues is dedicated to the challenging problem of how an RNA virus genome evolves and what are the primary mechanisms that drive the genetic changes. The authors used the cDNA copy of Citrus tristeza virus to inoculate N. benthamiana host and analyzed the DVG progeny of the master sequence – an approach that seems quite logical and adequate. This allowed to obtain novel data on the copy-choice recombination events upon CTV RNA replication and the hot spots for such events. The manuscript is well written and deserves publication is Viruses.
One point which might indicate a direction for the further research. It is known – as mentioned in this paper – that CTV has marked genetic stability despite the apparent luck of the proofreading mechanism described for replicases of large nidoviruses. However, the size of the closterovirus replicase genes is twice that of those in their distant relatives belonging to the Alpha-like virus superfamily. Is it possible that the unique protein domains in the expanded replicases of closterovirids influence the recombination pattern and the observed stability of their RNAs? Hopefully, these questions will find answers in future research.

Authors' response:

We thank the reviewer for the nice comments about our study and the manuscript as well as for the great suggestions regarding future research on unraveling the mechanism of the virus genetic stability.

Reviewer 3 Report

Comments and Suggestions for Authors

This is a very interesting paper that extends the knowledge of defective viral genomes. In this case, an infectious cDNA clone of CTV-T36 carrying GFP between p23 and the 3’ UTR was agroinfiltrated in Nicotiana benthamiana and viral-induced RNA sequences were examined by NGS in inoculated leaves and systemically infected leaves that developed after inoculation. This is a good experimental strategy, but I suggest the authors should mention that this system is a model system since it is not citrus. Moreover, I think the title should be “Intra-host citrus tristeza virus populations during prolonged infection initiated by a well-defined sequence variant in Nicotiana benthamiana”. It was good that much of the data produced agreed with the CTV-VT strain in citrus. My question is why this work was not done in citrus (e.g. Citrus macrophylla)? Granted, getting systemically infected citrus requires a lot of extra work and is more difficult.  My other question is why GFP was used in this experiment? What role did it play in these experiments?

Ln 16-22.  warrants that infection was initiated by a known well-defined sequence variant of the virus.? T36 limits single nucleotide mutants during infection yet the authors found four types of DVG’s. This sound like a lot. Clarify.

Ln 126 plan-to-plant passages…?

Ln 342. Table 3 is missing.

Author Response

Reviewer's comments and authors' response:

This is a very interesting paper that extends the knowledge of defective viral genomes. In this case, an infectious cDNA clone of CTV-T36 carrying GFP between p23 and the 3’ UTR was agroinfiltrated in Nicotiana benthamiana and viral-induced RNA sequences were examined by NGS in inoculated leaves and systemically infected leaves that developed after inoculation. This is a good experimental strategy, but I suggest the authors should mention that this system is a model system since it is not citrus. Moreover, I think the title should be “Intra-host citrus tristeza virus populations during prolonged infection initiated by a well-defined sequence variant in Nicotiana benthamiana”. It was good that much of the data produced agreed with the CTV-VT strain in citrus. My question is why this work was not done in citrus (e.g. Citrus macrophylla)? Granted, getting systemically infected citrus requires a lot of extra work and is more difficult.  My other question is why GFP was used in this experiment? What role did it play in these experiments?

We greatly appreciate the reviewer’s comments and believe that those helped to improve the manuscript.

As was suggested by the reviewer, we modified the title as: “Intra-host citrus tristeza virus populations during prolonged infection initiated by a well-defined sequence variant in Nicotiana benthamiana”.

There are few reasons for why this study was conducted in Nicotiana benthamiana. First, the production of defective RNAs upon CTV infection in citrus has been shown previously. With that, no such data were available for the CTV infection in N. benthamiana, a model host that is now being routinely used in CTV research. Furthermore, it was not known how stable is the CTV genome during virus infection of this host. Secondly, the use of N. benthamiana allowed us to initiate infection with a well-defined virus sequence derived from a viral cDNA clone, in contrast to inoculating citrus that is done using a population of viral genomes pre-accumulated in a source plant. Thirdly, in our experimental set up, we were able to compare the production of viral RNAs at the introduction site (initially inoculated leaves) and the systemically infected leaves. The use of the GFP gene inserted in the virus genome allowed us to visualize and monitor the development of infection and aided in collection of systemic leaves with comparable level of infection for further analysis.

We revised the manuscript according to the reviewer’s suggestions as shown below.

Ln 16-22.  warrants that infection was initiated by a known well-defined sequence variant of the virus.? T36 limits single nucleotide mutants during infection yet the authors found four types of DVG’s. This sound like a lot. Clarify.

We thank the reviewer for the comment. With this, we believe that the clarification on those statements provided in the downstream sections of the manuscript is sufficient for the readability of the manuscript.

Ln 126 plan-to-plant passages…?

Corrected as suggested by the reviewer

Ln 342. Table 3 is missing.

We apologize for Table 3 been missed in the previous version. Somehow, it got deleted during the formatting of the manuscript. We have inserted it in the revised version.
